# UNIVERSAL DISCRIMINATIVE QUANTUM NEURAL NET-WORKS

## ABSTRACT

Quantum mechanics fundamentally forbids deterministic discrimination of quantum states and processes. However, the ability to optimally distinguish various classes of quantum data is an important primitive in quantum information science. In this work, we trained near-term quantum circuits to classify data represented by quantum states using the Adam stochastic optimization algorithm. This is achieved by iterative interactions of a classical device with a quantum processor to discover the parameters of an unknown non-unitary quantum circuit. This circuit learns to simulate the unknown structure of a generalized quantum measurement, or positive-operator valued measure (POVM), that is required to optimally distinguish possible distributions of quantum inputs. Notably we used universal circuit topologies, with a theoretically motivated circuit design which guaranteed that our circuits can perform arbitrary input-output mappings. Our numerical simulations showed that quantum circuits could be trained to discriminate among various pure and mixed quantum states, exhibiting a trade-off between minimizing erroneous and inconclusive outcomes with comparable performance to theoretically optimal POVMs. We trained the circuit on different classes of quantum data and evaluated the generalization error on unseen quantum data. This generalization power hence distinguishes our work from standard circuit optimization and provides an example of quantum machine learning for a task that has inherently no classical analogue.

## 1 INTRODUCTION

Quantum computation are known to provide speedups in several applications over classical computation. Besides the famous Shor's algorithm for prime number factorization, quantum computers can also produce statistical patterns that are hard for a classical computer to produce. This raises the possibility that quantum computers can also recognize patterns that are hard to recognize for classical computers, or, in general, that quantum computers can help solve classical machine learning problems more efficiently. Recently, this joint field of quantum computation and machine learning has received a considerable amount of attention. Using the circuit model of computation, several quantum algorithms have been designed that provide quadratic to exponential speedups on classical data (Biamonte et al. (2017); Ciliberto et al. (2018)).

A related and interesting area is to develop novel machine learning methods on quantum data. In general, any collection of quantum states which carry certain meaningful information can be considered as quantum data. To motivate this direction, we want to emphasize that using quantum states as a storage medium for information processing have shown to provide advantages in several aspects. Researches have shown that by coupling a quantum state with another target system, one could obtain information about the target system through this quantum state but with a better sensitivity. For example, quantum meteorology allows for a quadratic improvement over classical methods in terms of the scaling of statistical errors, i.e. the scaling of the standard deviations in estimated values in repeated measurements. Another example is quantum sensing, which provide much higher sensitivity in tasks like target detections in microwaves, i.e. quantum radar (Barzanjeh et al. (2015)), and in general, sensing electric or magnetic fields (Degen et al. (2017)). Further, it is proposed that by storing certain information of a picture on a quantum state, pattern recognition can be achieved with

much fewer damage done to this picture if it is very sensitive to the exposure of light (Schaller & Schützhold (2006)).

Certain types of quantum data are special in that they are inherently quantum mechanical. They could be produced from materials fabricated in certain exotic quantum state or could be the result of quantum information processing procedures. Here we could conjecture the inherent advantage of using quantum computers to recognize and classify these natural quantum data. For example, topological materials made in the exotic topological phase have non-classical electronic properties and are promising materials to build fault-tolerant quantum computers (Qi & Zhang (2011); Karzig et al. (2017)). However, predicting the phase of topological materials has been a very challenging problem, but recently it was shown that quantum neural networks could be used to recognize the phase of a quantum state (Cong et al. (2018)). In addition, the promised security of quantum communication protocols and a surge of ideas in quantum communication networks (Kimble (2008); Ren et al. (2017)) further stimulates the research into the topic of quantum data.

In this work, we explored the general problem of classifying quantum data. This problem could be considered as an extension of the well-researched field *quantum state discrimination*, which identifies a quantum state among a set of possible candidates of which one has complete knowledge *a priori*. Quantum state discrimination already found applications in many fields such as quantum key distribution (Bennett (1992a)) and cryptography (Barnett & Croke (2009); Bergou (2007)). A key challenge for the discrimination of quantum states is that a deterministic discrimination is impossible when the complex vectors representing the candidate states are not orthogonal, i.e. when their overlaps are non-zero. Therefore, the central task of quantum state discrimination is to find the optimal discriminative measurement that one should perform for those states.

However, it is not possible to directly apply quantum state discrimination to classify the quantum data. Firstly, it is inappropriate to assume that one possesses the complete knowledge of the data *a priori*, which are often only samples generated from the data collection process. Also, even with all the quantum data available, obtaining the complete knowledge of them (i.e. the density matrices describing them) are very expensive. In addition, *quantum state discrimination* often fails to give the optimal discriminative measurement in an analytically closed form, unless the quantum states are already orthogonal or possess certain symmetry properties (Barnett & Croke (2009)). In cases it fails, one may use numerical optimization algorithms to find the optimal measurement. However, the exponential increase of the dimension of the density matrices renders the numerical optimization inefficient.

Due to the limitations of quantum state discrimination, it is then a natural question whether we can use a quantum computer to help with the optimization procedure. Here we utilize a hybrid quantum-classical approach to directly learn the design of a shallow quantum circuit for the classification of quantum data. In this hybrid scheme, a classical computer is used to interactively change a quantum circuit and optimize the output of the quantum computer. Recently a string of works focusing on this hybrid scheme for training quantum circuits for a wide range of tasks have been proposed (Banchi et al. (2016); Wan et al. (2017); Innocenti et al. (2018); Romero et al. (2017); Mitarai et al. (2018); Farhi & Neven (2018); Verdon et al. (2017); Li & Benjamin (2017); Grant et al. (2018); Schuld et al. (2018)), with major players in quantum computing hardware releasing software toolkits for hybrid quantum-classical models. This makes the hybrid scheme a promising area in quantum computation.

Our approach described here is novel in two perspectives. First, we used a quantum circuit ansatz that is motivated for implementation on near-term devices. This ansatz is shallow in depths but possess the full power of quantum computation, i.e. it can perform any unitary transformation allowed in quantum mechanics. It comprises gates from a universal gate set consisting of C-NOT and single-qubit gates, motivated by the fact that their implementations are known for the currently most used experimental architectures. It is nearly optimal in terms of the number of C-NOT gates it has, which is also an important feature for an implementation on near-term devices. Second, unlike previous works on quantum state discrimination, we focused on the generalization ability of our circuit, i.e., we trained the circuit on a specific range of the parameters with the goal of maximizing its generalization performance, hence considering a learning framework. This distinguishes our work from the pure optimization problem for the state discrimination task, which is optimizing the circuit to distinguish only a concrete set of states. We showed that this universal quantum circuit can be trained as a discriminator for classification of non-orthogonal quantum data sampled from

various different probability distributions. Our discriminator can achieve a near-zero error rate by producing inconclusive signals.

## 2 DATASET AND OUR APPROACH

In this work, we propose a novel approach to train a universal quantum circuit to classify quantum data, which are stored in qubits. In this section, we first explain the mathematical descriptions of quantum states, operations and measurements on these states. We then specify the quantum data we used in this work for classification. Next, we outline the approach we took to optimize a universal quantum circuit which was used to classify the quantum data. We defer the detailed decomposition of this quantum circuit in Appendix A.

**Mathematical descriptions**  Quantum states are described by density matrices, which are Hermitian, positive semi-definite complex matrices of unit trace, and commonly denoted by the symbol $\rho$. For the quantum states which stores certain useful information, we may assume that they could be parameterized by certain parameters which follow certain probability distributions specific to the information they carried. For classification, we are normally presented with an unknown quantum state, belonging to families of quantum states, each described mathematically by:

$$\rho_i(a_i),\ a_i \sim \alpha_i, \tag{2.1}$$

where $i$ is the label for different families, $\rho_i(a_i)$ is the density matrix describing a quantum datum in the family $i$, parametrized by $a_i$. The parameters $a_i$ are assumed to follow the probability distribution $\alpha_i$. When training a quantum circuit for classification, $a_i$ are samples drawn according to the distribution $\alpha_i$.

Transformations on quantum states are described by a complex unitary matrix $U$, which transforms a quantum state $\rho$ according to the rule:

$$\rho \to U\rho U^\dagger. \tag{2.2}$$

A measurement on the quantum state $\rho$ is described by a set of matrices $\{M_j\}$, which are Hermitian, positive semi-definite and sum to the identity matrix, i.e. $\sum_j M_j = \mathbb{1}$. Here $j$ labels the possible measurement outcomes, and the probability $p_j$ that this particular measurement outcome is detected is,

$$p_j = \mathrm{Tr}(M_j\rho), \tag{2.3}$$

where Tr is the trace operator. Such a collection of matrices $M_j$ is commonly called a *positive-operator valued measure* (POVM). A common example of POVM is a *projection-valued measure* (PVM). In the case of a PVM, each $M_j$ is a projector into some linear subspace and different $M_j$ are orthogonal to each other, i.e. $M_j M_i = \delta_{ij} M_j$. Any POVM could be realized by a quantum circuit, which consists of a series of unitary matrices (transformations) and measurements in the computational basis. Conversely, a quantum circuit which is parameterized by some parameters and has measurements could also represent many different POVMs. There exists a quantum circuit which could represent any POVM with a fixed number of possible measurement outcomes. Such a circuit is called a universal discriminator in this paper, and a specific one which we used is discussed in Appendix A.

**Our Dataset**  For this work, we restricted our attention to the classification of two families of quantum states stored in 2 qubits. Our first family consists of pure states, parametrized by a real number $a \in [0,1]$:

$$\psi_1(a) = \left(\sqrt{1-a^2}, 0, a, 0\right),\ \rho_1 = |\psi_1(a)\rangle\langle\psi_1(a)|. \tag{2.4}$$

The second family consists of mixed states $\rho_2(b)$ where $b \in [0,1]$. Specifically,

$$\psi_{2/3} = \left(0, \pm\sqrt{1-b^2}, b, 0\right), \rho_2 = \frac{1}{2}\left|\psi_2\rangle\langle\psi_2\right| + \frac{1}{2}\left|\psi_3\rangle\langle\psi_3\right|. \tag{2.5}$$

The overlap between $\psi_1$ and $\psi_{2/3}$ is $ab$, and hence our data in two families are non-orthogonal. In the case when $a$ is a fixed value, and $b = \frac{1}{\sqrt{2}}$, the maximal success rate for unambiguously discriminating between $\rho_1$ and $\rho_2$ has been theoretically studied, and an experimental demonstration is available (Mohseni et al. (2004)). The specific distribution we have tested in our experiments are summarized in Table 1. To generate the data for the training, validation, and testing of our circuits, we randomly sampled points from the corresponding distributions.

| | Family 1 ($\rho_1(a)$) | Family 2 ($\rho_2(b)$) |
|---|---|---|
| Case 1 | $a \in [0,1]$ | $b \in [0,1]$ |
| Case 2 | $a \approx 0.25$ | $b \approx \frac{1}{\sqrt{2}}$ |
| Case 3 | $a \approx 0.25$ | $b \in [0,1]$ |
| Case 4 | $a \in [0,1]$ | $b \approx \frac{1}{\sqrt{2}}$ |

Table 1: A summary of different test cases we classified in this work. Here $a(b) \in [0,1]$ represents that $a(b)$ follows a uniform distribution in $[0,1]$. $a(b) \approx 0.25(\frac{1}{\sqrt{2}})$ represents that $a(b)$ follows a normal distribution with mean $0.25(\frac{1}{\sqrt{2}})$, standard deviation $0.05$, and which is truncated in $[0,1]$.

**Approach.** Overall, there are two major strategies to cope with our inherent inability to perform deterministic discrimination of quantum states: (a) *Minimum-error discrimination*: In this strategy, the task is to minimize the probability that the inevitable errors occur in the classification. (b) *Unambiguous discrimination*: In this strategy, the discriminator has one more output prediction than the number of classes it tries to classify: an inconclusive outcome. The task is to eliminate the error rate of the discriminator while minimizing the probability of this inevitable inconclusive outcome. A pure unambiguous discrimination with strictly zero error rate is not guaranteed to be possible for arbitrary quantum data. From the perspective of numerical optimization, one needs to allow for some small but non-zero errors to happen.

In this work, we used the machine learning approach to train a universal quantum circuit capable of giving any quantum measurements with four possible measurement outcomes $m_{i_2 i_1}$, where $i_1, i_2 \in \{0,1\}$ are the measurement outcomes of the first and the second qubit respectively. The parameterization of this circuit is discussed in Appendix A. By assuming that input $\rho_1(a)$ produces the output $m_{00}$ or $m_{10}$, input $\rho_2$ produces the output $m_{01}$, and assuming that $m_{11}$ is the inconclusive output, this circuit acts as a discriminator. Therefore, we could trivially define various probabilities (success probability $P_{\text{suc}}$, error probability $P_{\text{err}}$, and inconclusive probability $P_{\text{inc}}$) with respect to an input training data with known class label. For example, when $\rho_1$ is the input, the probability of detecting $m_{01}$ is the $P_{\text{err}}$, the probability of detecting $m_{11}$ is the $P_{\text{inc}}$. In this work, we performed experiments on simulated quantum computers, where these probabilities were available. We note that on real quantum computers, these probabilities can be estimated through repeated measurements on replicated data, up to some precision.

To train the circuit, we used a heuristically motivated loss function defined in Eq. 2.6, which is the averaged absolute difference between the desired probabilities and the measured probabilities. It contains hyperparameters $\alpha_{\text{err}}$ and $\alpha_{\text{inc}}$ to balance between the erroneous outcomes and the inconclusive outcomes:

$$J = \sum_i \frac{1}{|S_i|} \sum_{a_i \in S_i} |P_{\text{suc}}(\rho_i(a_i)) - 1|$$
$$+ \alpha_{\text{err}} \sum_i \frac{1}{|S_i|} \sum_{a_i \in S_i} |P_{\text{err}}(\rho_i(a_i)) - 0|$$
$$+ \alpha_{\text{inc}} \sum_i \frac{1}{|S_i|} \sum_{a_i \in S_i} |P_{\text{inc}}(\rho_i(a_i)) - 0|. \tag{2.6}$$

Here we assumed that for each family of quantum states, we had been supplied with a set $S_i$ of training samples, where each class was labeled by $i$. Then $|S_i|$ denotes the number of samples in the training sets $S_i$, $\alpha_{\text{err}}$ is the penalty for making errors, and $\alpha_{\text{inc}}$ is the penalty for giving inconclusive outcomes. $P_{\text{suc}}(\psi)/P_{\text{err}}(\psi)/P_{\text{inc}}(\psi)$ are the probabilities of giving a correct/erroneous/inconclusive measurement outcome for the specific input quantum data $\rho_i$. This loss function measures the performance of our quantum circuit as a minimal-error discriminator (when $\alpha_{\text{err}} < \alpha_{\text{inc}}$) or as an unambiguous discriminator (when $\alpha_{\text{err}} > \alpha_{\text{inc}}$).

To train this circuit, we used the Adam optimization algorithm (Kingma & Ba (2014)), and the gradients were calculated by the forward difference formula.

For our specific problem of classifying $\rho_1$ and $\rho_2$ as defined in Eq. 2.4 and Eq. 2.5, we defined an extra set of success/erroneous/inconclusive rates in Eq. 2.7 to summarize and compare the performance of different trainings:

$$\begin{aligned} P_s &= \frac{1}{3}P_s(\rho_1)_{\text{avg}} + \frac{2}{3}P_s(\rho_2)_{\text{avg}} \\ &= \frac{1}{3}P_s(\psi_1)_{\text{avg}} + \frac{1}{3}P_s(\psi_2)_{\text{avg}} + \frac{1}{3}P_s(\psi_3)_{\text{avg}} \end{aligned} \tag{2.7}$$

where $s$ stands for *suc* (successful), *err* (erroneous) or *inc* (inconclusive). The subscript *avg* means that the probabilities are calculated as the value averaged in all samples available in either the training set, or the test set (but not both). The choice of weights ($\frac{1}{3}$ and $\frac{2}{3}$) in the Eq. 2.7 was made to be consistent with the paper Mohseni et al. (2004).

## 3 THEORETICAL ANALYSIS

Here we describe a theoretical result to which we will compare our numerical results. First, we mention a general result. Assume we have a family of quantum data $\rho(a)$, each one is parameterized by $a$ and occur with a probability $P(a)$. Assume in addition that we have a quantum measurement described by a POVM with elements $\{\Pi_i\}_{i \in \mathbb{N}}$, where $i$ labels different measurement outcomes. Then, the probability of detecting measurement outcome $i$, averaged over any possible quantum data $\rho(a)$, is,

$$\int \text{Tr}(\Pi_i \rho(a))P(a)\mathrm{d}a = \text{Tr}\left[\int \Pi_i \rho(a)P(a)\mathrm{d}a\right] = \text{Tr}\left[\Pi_i \int \rho(a)P(a)\mathrm{d}a\right]$$
$$= \text{Tr}[\Pi_i \rho], \tag{3.1}$$

where $\rho \equiv \int \rho(a)P(a)\mathrm{d}a$, and the integration of the matrix is done in an element-wise fashion. Therefore, if $\text{Tr}(\Pi_i \rho) = 0$ for some $i$, then $\int_D \text{Tr}(\Pi_i \rho(a))P(a) = 0$ for any subset $D$ with non-zero measure in the whole parameter space of $a$. This is due to the fact that $\text{Tr}[\Pi_i \rho(a)]P(a) \geq 0$ for any parameter $a$.

Applied to the problem of unambiguous discrimination, it is obvious that the problem of unambiguously discriminating $\rho_1 = \int_a \rho_1(a)P_1(a)\mathrm{d}a$ and $\rho_2 = \int_a \rho_2(a)P_2(a)\mathrm{d}b$, is equivalent to the problem of unambiguously discriminating the family $\rho_1(a), \forall a$, from the family $\rho_2(b), \forall b$, where $P_1(a)/P_2(b)$ is the probability of occurrence of $\rho_1(a)/\rho_2(b)$. That is, if $\{\Pi_1, \Pi_2, \Pi_?\}$ is a POVM that unambiguously classifies all members of the two families $\rho_1(a)$ and $\rho_2(b)$, for all possible parameters, i.e. $\Pi_?$ corresponds to the inconclusive outcome with,

$$\text{Tr}(\Pi_2 \rho_1(a)) = 0, \ \forall a,$$
$$\text{Tr}(\Pi_1 \rho_2(b)) = 0, \ \forall b,$$

then $\text{Tr}(\Pi_1 \rho_2) = \text{Tr}(\Pi_2 \rho_1) = 0$, and vice versa. Using this formalism, we theoretically analyzed the different cases we described in Table 1 based on the works of Raynal et al. (2003) and Barnett & Croke (2009), and the results are displayed in Table 2. Note that these are average case success probabilities.

| | Family 1 ($\rho_1$) | Family 2 ($\rho_2$) | $P_{\text{suc}}$ |
|---|---|---|---|
| Case 1 | $a \in [0, 1]$ | $b \in [0, 1]$ | 0.67 |
| Case 2 | $a \approx 0.25$ | $b \approx \frac{1}{\sqrt{2}}$ | 0.64 |
| Case 3 | $a \approx 0.25$ | $b \in [0, 1]$ | 0.76 |
| Case 4 | $a \in [0, 1]$ | $b \approx \frac{1}{\sqrt{2}}$ | 0.55 |

Table 2: A summary of maximal success rate when the error rate is exactly 0 for different test cases we have classified in this work. They were calculated using Eq. 2.7 and obtained theoretically using methods mentioned in Section 3.

## 4 NUMERICAL RESULTS ON SIMULATED QUANTUM COMPUTERS

In this work, we aim to train a universal discriminator to discriminate different families of quantum data. Here, we presented the results of training the universal discriminator to discriminate different distributions summarized in Table 1 on a simulated quantum computer. To balance between eliminating the error rate ($P_{\text{err}}$) while minimizing the inconclusive rate ($P_{\text{inc}}$), we used a specific training strategy described in the following. We first prioritized a smaller inconclusive rate by starting with a zero penalty for erroneous outcomes ($\alpha_{\text{inc}} > \alpha_{\text{err}} = 0$), and then increased the $\alpha_{\text{err}}$ in a step-wise manner until a certain objective error rate is achieved. Similar optimization procedures have been used in the context of variational auto-encoders both in classical machine learning (Sønderby et al. (2016)), and in quantum machine learning applications (Rocchetto et al. (2017)). Using this scheme, we trained our circuit to unambiguously discriminate the two families of quantum states and observed a convergence towards the theoretical success rates obtained in Section 3 with increasing amount of data used to train the discriminator . Notably, we did not observe any signs of overfitting despite the varying size of the training dataset (Figure 1 a).

### 4.1 TRADE-OFF BETWEEN THE ERROR RATE AND THE INCONCLUSIVE RATE

In another test, we showed that our model is able to obtain a much higher success rate ($P_{\text{suc}}$) if we allow a slightly higher error rate compared with previous results during the trainings. This hints at a trade-off between the error rate ($P_{err}$) and the inconclusive rate ($P_{inc}$) which can be utilized in real-world applications.

Specifically, for the dataset "Case 4" in Table 1 we fixed the two penalties, $\alpha_{\text{err}}$ and $\alpha_{\text{inc}}$, during the training and observed a gradual transition from unambiguous-like classification (characterized by a near-zero error probability) to minimal-error-like classification (characterized by the near-zero inconclusiveness) when we used varying penalties (Figure 2 a-c) throughout the different trainings. Allowing a small error rate resulted in a much higher success rate, which has not been predicted theoretically. We note that introducing the penalty terms $\alpha_{\text{err}}$ and $\alpha_{\text{inc}}$ made the training process also more stable (Figure 2 a). Therefore, the hyperparameters $\alpha_{\text{err}}$ and $\alpha_{\text{inc}}$ acted as a form of regularization and could be adjusted to give a higher success probability or a lower inconclusiveness rate for the final model (Figure 2).

For all the datasets in Table 1, we achieved a much higher success rate than the theoretical case of an exactly zero error rate, with the required error rate tolerance smaller than 0.01 (Figure 3).

## 5 LEARNING CONVERGENCE FROM ENSEMBLE MEASUREMENTS

Although we trained our circuit using probabilities which are not available on an actual quantum device, we can estimate these probabilities from repeated measurements on this quantum device. We found that when using probabilities estimated from repeated measurements of quantum states, the training process still converged in all cases. In practice, the noise introduced by the inferred probabilities could be effectively countered by increasing the number of repeated measurements, using a lowered error rate, and adjusting the step size for gradient calculation using the forward difference formula. Overall, a combination of a step size of $0.01$, $10^5$ repeated measurements, and a learning rate of $10^{-3}$ well approximated the result obtained using exact probabilities. Therefore, our study here is feasible for actual quantum devices. We leave open the actual implementation as future projects. Further discussions are deferred in Appendix B.

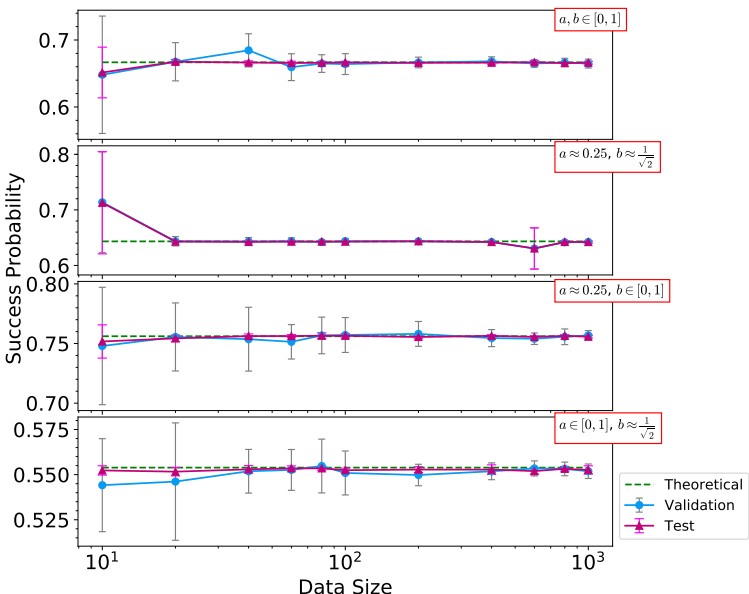

(a) The success rates converged to the theoretical result (the green dashed line in the figure) as the training data size was increased. The performance on the testing (purple line) and the validation dataset also converged with the growing data size. The abnormal deviation at the point with data size 600 for the case $a \approx 0.25$ and $b \approx 1/\sqrt{2}$ was due to unsuccessful optimizations.

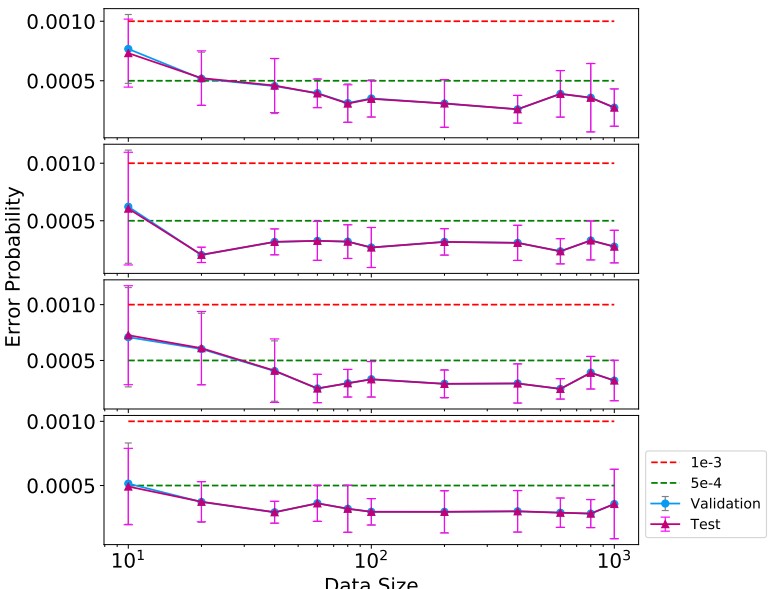

(b) Most of the error rate for the corresponding trainings are well below 0.0005, indicating an unambiguous discrimination was achieved.

Figure 1: Unambiguous classification of non-orthogonal quantum data sampled from different probability distributions. The data were averaged over 10 repeated trails starting with random initializations and the bars indicate the standard deviations. The training, validation and test data were sampled from the corresponding distributions. The data size indicates the size of the training and the validation set. The test data was fixed at a size of $10^4$ for each family and each distribution.

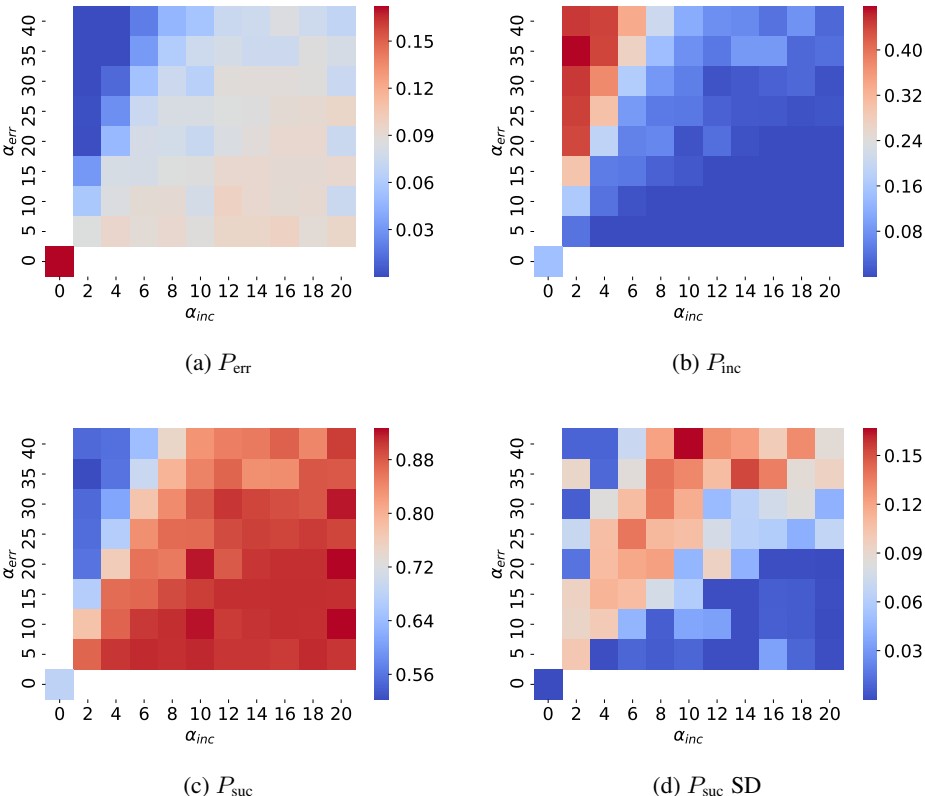

Figure 2: **With different penalties we observed a trade-off between the error rate and the in-conclusive rate.** Compared with the point $\alpha_{err} = \alpha_{inc} = 0$ (bottom left corner), the added penalties improved the success probability or the inconclusiveness respectively. **(a)-(c)**: The gradual transition from the unambiguous classification (near-zero error rate, top left corner) to a minimal error classification (near-zero inconclusiveness, bottom right corner) with changes in the error penalty $\alpha_{err}$ and the inconclusiveness penalty $\alpha_{inc}$. We observed that the gain in the success rate was around 0.32 when we made a sacrifice of only 0.1 in the error rate. The data was tested on $a \in [0, 1]$, and averaged over 50 repeated trails with random initializations. **(d)**: Standard deviation for $P_{suc}$. With an increasing standard deviation (closer to the diagonal line), the result became increasingly unstable when the two penalties ($\alpha_{err}$ and $\alpha_{inc}$) are closer in value. The standard deviations for $P_{err}$ and $P_{inc}$ showed the same pattern as for $P_{suc}$ (not shown).

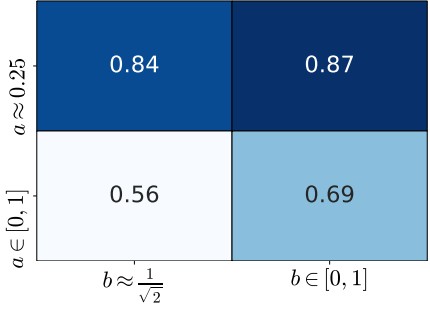 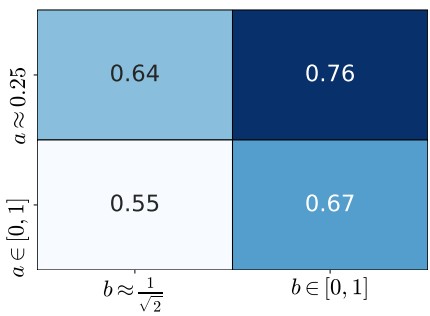

(a) Success rate for trained circuit for classification of data

(b) The theoretical success rate when exact 0 error rate is achieved.

Figure 3: **Unambiguous discrimination of data sampled from different probability distributions with higher success rate.** **(a)** Trained quantum circuits were capable of classifying quantum data which was sampled from a variety of different mixed probability distributions for $\rho_1(a)$ and $\rho_2(b)$. The classification was done in an unambiguous manner (with error rate $< 0.01$). **(b)** For comparison, we included here the theoretical result mentioned in Table 2.

## 6 CONCLUSIONS

We have developed a universal quantum circuit learning approach for the classification of quantum data. In particular, we have designed a theoretically motivated loss function and used the stochastic optimization algorithm Adam in a quantum-classical hybrid scheme to train a circuit to perform quantum state discrimination. This training process generalized well for the discrimination task on new data, i.e., states from the parameter range which have not been seen during the training process. This in particular distinguishes our work from previous results on quantum circuit learning, in particular the very recent study in Fanizza et al. (2018), which only optimizes circuits for specific inputs. Note that this prior work hence does not consider the generalization ability and hence does not treat the actual learning problem, which aims at optimization as well as generalization.

In our work, we observed a trade-off between the error rates and the inconclusive rates when we penalized them differently in the loss function. Although this experiment was done on simulated quantum computers where exact measurement probabilities are available, we showed that this optimization could be experimentally performed with repeated measurements of the quantum states. Finally, we note that the recent quantum methods for estimating the analytical gradient via variations in the unitaries (Mitarai et al. (2018)) can be directly applied to training our circuits and therefore one can perform the optimization efficiently on near-term quantum devices.

With the progress on technologies for preservation and transportation of quantum states, we can expect many applications of a trained discriminative quantum circuits introduced here. Quantum state discrimination by itself plays a key role in quantum information processing protocols and is used in quantum cryptography (Bennett (1992b)), quantum cloning (Duan & Guo (1998)), quantum state separation, and entanglement concentration (Chefles (2000)). Our work can provide improvements on these traditional areas by producing a classifier that is resilient to the statistical noise found in the actual communication. For example, we can consider an improved version of the B92 quantum key distribution protocol (Bennett (1992a)) by including the noise-induced randomness in its two quantum keys and classify them with our discriminative circuit. Furthermore, we can consider training a discriminative quantum circuit used to construct quantum repeaters and state purification units within quantum communication networks. The training can take quantum data that have noise specific to the communication networks and therefore produces a discriminator that can recognize and filter those noise to provide better performance. Our discriminator can also be used to verify the output of other generative models, such as the quantum version of Boltzmann machines (Amin et al. (2018)), or generative artificial neural networks (Goodfellow et al. (2014); Lloyd & Weedbrook (2018)).

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

# A QUANTUM CIRCUITS FOR POVM

This section describes the parametrization of the circuit capable of performing any quantum measurement on 2 qubit inputs with 4 possible measurement outcomes. This circuit could be represented by the following circuit diagram:

$$|0\rangle \quad /^2 \quad \boxed{V} \quad \boxed{\measuredangle} \qquad M \tag{A.1}$$
$$|\psi\rangle \quad /^2$$

## A.1 COSINE-SINE DECOMPOSITION

Here we mention the cosine-sine decomposition of unitary matrices, which will be frequently used in the following sections. For every unitary matrix $U \in \mathbb{C}^{2^n \times 2^n}$, it can be decomposed as:

$$U_n = \begin{pmatrix} A_0 & 0 \\ 0 & A_1 \end{pmatrix} \begin{pmatrix} C & -S \\ S & C \end{pmatrix} \begin{pmatrix} B_0 & 0 \\ 0 & B_1 \end{pmatrix} \tag{A.2}$$

where $A_0, A_1, B_0, B_1$ are unitary matrices of size $2^{n-1} \times 2^{n-1}$, $C$ and $S$ are real diagonal matrices of size $2^{n-1} \times 2^{n-1}$ satisfying $C^2 + S^2 = \mathbb{1}$. It can be written in the following circuit equivalence diagram:

$$n-1 \left\{ \boxed{U_n} \right\} = \left\{ \boxed{U_{n-1}} - \boxed{\phantom{x}} - \boxed{R_y} - \boxed{\phantom{x}} \\ \boxed{U_{n-1}} \right. \tag{A.3}$$

Here a box $\square$ represents the control part of a uniformly controlled gate, see section IV of Iten et al. (2015) for details. In the circuit in Fig. A.1, the first qubit is initiated to be $|0\rangle$, so we have:

$$|0\rangle \quad \boxed{U_n} \\ n-1 \quad = \quad \boxed{U_{n-1}} - \boxed{R_y} - \boxed{\phantom{x}} - \boxed{U_{n-1}} \tag{A.4}$$

## A.2 DECOMPOSITION OF CIRCUIT IN FIG. A.1

For a general measurement giving at most 4 measurement outcomes, we have the following circuit representation:

$$|0\rangle \quad \boxed{\phantom{V}} - (M_1) \\ |0\rangle \quad \boxed{V} - (M_2) \\ |\phi\rangle \\ |\psi\rangle \tag{A.5}$$

The first $V$ could be decomposed using the circuit equivalence on page 5 of Iten et al. (2016) into:

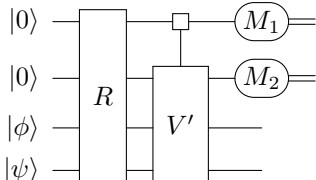

where the $R$ gate does not act on the second qubit. Applying the cosine-sine decomposition gives:

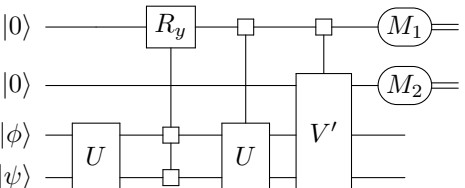

The uniformly controlled $V'$ and $U$ can be merged and put after the measurement of $M_1$ as:

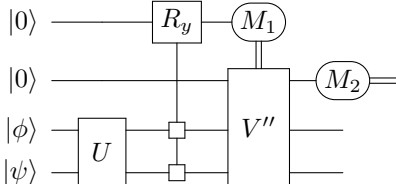

The first line of the circuit could be merged with the second line as follows:

(A.6)

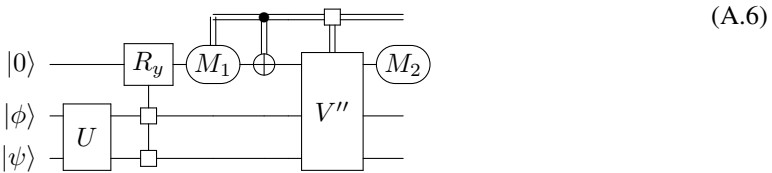

And then we can apply the cosine-sine decomposition to $V''$. Throwing away the last gate on the third and the fourth qubits, we obtain:

(A.7)

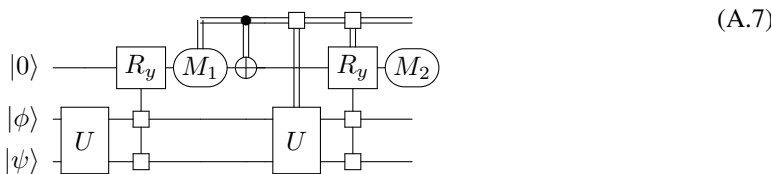

The uniformly-controlled rotations and the remaining two-qubit unitary gates could be easily parametrised by CNOTs and single qubit rotations. For example, see Shende et al. (2006) and Shende et al. (2004).

## B    LEARNING CONVERGENCE FROM ENSEMBLE MEASUREMENTS

Here we simulate the process that a classical-quantum hybrid scheme would implement utilizing a quantum device and analyse its performance. These numerical simulations can in principle be validated in a physical experiment, where the measurement outcomes are used to infer the different probabilities for the cost function. To have a good estimation of the probabilities, and hence the cost function, one has to make repeated measurements to train the model, and we note that in particular better methods to evaluate the analytical gradient are available on a shallow quantum device (Mitarai et al. (2018)). We first give a brief discussion of the estimated number of repeated measurements which are required to approximate the gradient. This follows the treatment of Farhi & Neven (2018)[Section 3]. Since the gradients are calculated using the forward difference formula:

$$\frac{df}{dx}(x) = \frac{f(x+\varepsilon) - f(x)}{\varepsilon} + O(\varepsilon) \qquad (B.1)$$

The error in the calculation of $f$ must be at most of the order of $O(\varepsilon^2)$, in order to prevent dominating the total error. To achieve this ideally with a $99\%$ probability, one requires the number of repeated measurement to be of the order $\frac{1}{(\varepsilon^2)^2} = \frac{1}{\varepsilon^4}$.[1] For example, when $\varepsilon = 10^{-3}$, the ideal number of repetitions is given by $10^{12}$.

In practice, we do not use $\frac{1}{\varepsilon^4}$ measurements, since the Adam optimization algorithm is designed with the noise of the cost function taken into account. To give an estimate of the number of repeated measurements which are required for the convergence of the optimization process, we performed two numerical experiments. We first looked at the case when the number of repeated measurements

---

[1] This assumes that the cost function follows a normal distribution with variance of the order $\frac{1}{\sqrt{N}}$, where $N$ is the number of measurements made in reach run in order to calculate the cost function.

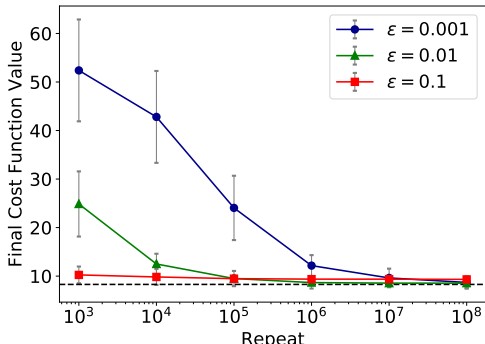

Figure 4: The cost function after 5000 iterations. The result obtained using exact probabilities is shown by the horizontal dashed line. For a smaller step size ($\varepsilon$) for gradient calculations, we found that more repetitions were required to give a consistent result. However, a combination of $\varepsilon = 10^{-2}$ and $10^5$ repetitions gave a result which well approximated the result obtained using exact probabilities. Here *repeat* is the number of repeated measurements that were made each time to calculate the cost function. The cost function values were averaged over $50$ repeated runs of the training process, and the bars indicate the standard deviations.

was large ($\geq 10^3$) and $\varepsilon = 10^{-2}$. We found that $10^5$ repeated measurements for each iteration were a robust configuration for a successful convergence. Second, we used a small number of repeated measurements but varied the learning rate and increased the maximal number of iterations for Adam. Setting $\varepsilon = 10^{-2}$ and taking only $100$ repeated measurements, we observed that the optimizations were successful with a large number of iterations. In both experiments, the penalties were set to $\alpha_{inc} = 5$ and $\alpha_{err} = 40$.

**Large number of repetitions.** Our results showed that for a fixed maximum number of iterations (5000) for Adam, a combination of $\varepsilon = 10^{-2}$ and $10^5$ repeated measurements gave robust results, i.e., the final cost function was close to the value obtained with the exact probabilities (with an error within $3\%$) and was stable (with a relative standard deviation of $13\%$). A more detailed description of the trade-off between repeated measurements and the stability of the cost function is shown in Fig. 4.

**Small learning rates and high number of iterations.** Our numerical experiments further showed that in the case of using a small number of repeated measurements, lowered learning rates could effectively counter the noisy brought by the insufficient sampling. Although in this case, the optimization required a large number of iterations to finish. For example, with only $100$ repeated measurements, the variance of cost function $J_1$ after 20000 iterations decreased as we lowered the learning rate (Fig.5(a)). We could visually observe the optimization process where the cost function $J_1$ slowly approached the optimal value in Fig 5(b). Here, gradient step was taken as $\varepsilon = 10^{-2}$.

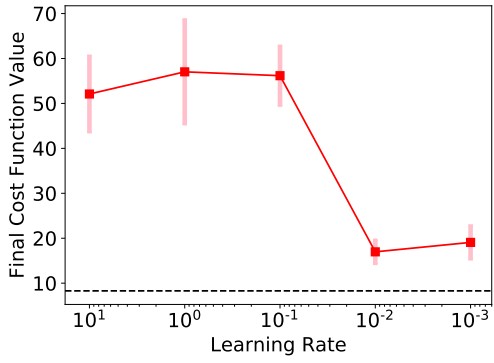 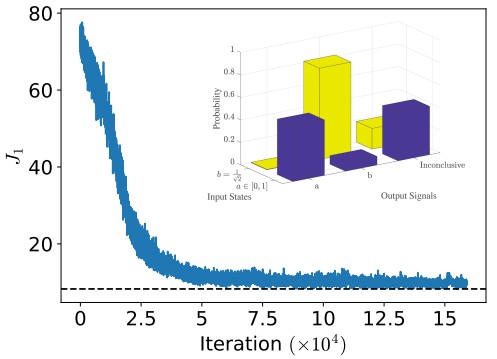

(a) Lowering learning rate to counter the effect of insufficient sampling. Both the value and the standard deviation of cost function at the 20000'th iteration were brought down by lowered learning rate. Here the number of repeated measurements was only 100, much smaller than $\frac{1}{\varepsilon^4} = 10^8$.

(b) The noisy cost function $J_1$ estimated by 100 repeated measurements slowly moved to its optimal value when the learning rate was set as $0.001$. The horizontal dashed line showed the minimal value $8.3$ for the cost function. The inset illustrates that trained circuit could discriminate the two quantum states. Although the error rate in the inset is not 0, we believe it could be achieved by further tuning the penalties, which was set as ($\alpha_{err} = 40$, $\alpha_{inc} = 5$) for this training.

Figure 5: Small learning rates with high number of iterations.

