# OpenReview forum: "Universal discriminative quantum neural networks"
_ICLR.cc/2019/Conference_

### Official Review · AnonReviewer1 · 2018-11-02
**Not double blind**

**Rating:** 2
**Confidence:** 2

**Review:**

Unfortunately, while this is interesting work, the authors emails are listed on the first page and the acknowledgments are very revealing. I am a big fan of Google, UCL, and the Royal Society, and this strongly biases my view of the work.

My biased review:

- the paper is interesting, and should go to another venue. I do not think the authors will get benefit from presenting this work at ICLR (there is a tiny quantum focus).

- how is the cost function justified? I'd be curious to see how the authors derived it. Right now above Eq 2.4 it seems like it is heuristic to balance successful/erroneous/inconclusive rates. If it is a heuristic, the paper should clearly state this.

- using simple examples of quantum data and quantum states would go a long way towards helping me understand the problem setup (Eq 2.1). It took me a while to grok this.

- The acronym POVM is never defined.

---

### Official Review · AnonReviewer2 · 2018-11-03
**Classifying a quantum state with a neural network--Not sure ICLR has the right audience for this**

**Rating:** 5
**Confidence:** 2

**Review:**

Summary of paper:
The authors partially integrate a neural network into classical approaches to classify the state of a quantum circuit. The model is not actually clear in what it is doing, but there are some trained weights somewhere. They allow for an "uncertainty" prediction by giving one more node than there are classification targets, corresponding to a less-penalized uncertain prediction. They evaluate their model on numerical simulations.

Strengths:
-
Weaknesses:
- The neural network architecture is entirely standard with nothing new.
- The paper is poorly written and very hard to follow.
- The focus is almost exclusively on the application, and yet the application is not explained effectively.
- The implication of the results and usefulness is not elaborated.
- The particular contributions are not clear.

Suggested Revisions:
- What is the 9b8d in the first sentence of the abstract?
- "...been developed to address the question [of] whether quantum mechanics..."
- "...for all the dataset[s] in Table 1..."

---

### Official Review · AnonReviewer3 · 2018-11-03
**Review TLDR: Ok good fit for ICLR maybe even better for QIP**

**Rating:** 5
**Confidence:** 3

**Review:**

Authors give a method to perform a full quantum problem of classifying unknown mixed quantum states. This is an important topic but the paper is ok and I think the test case is a bit lacking.

The theory  is sound and the math is good. The only question I have is how does this hold on a real quantum computer such as IBMQ/rigetti quantum computing etc.. or even under a noisy simulator

Although the paper is sounds and it is a good idea, the presentation is a bit lacking. There are several typos and formatting problems, such as excess spaces and some sort of hex code (9b8d) in the abstract which I am guessing is left over from the NIPS template.
Two other things is that usually in double blind review one should not leave the emails with affiliation and one should anonymize the Acknowledgements as well.

---

### Meta-Review · Area_Chair1 · 2018-12-12
**Paper needs improvement**

**Confidence:** 5
**Recommendation:** Reject

**Metareview:**

The paper needs work to improve clarity and strengthen the technical message. Also, the authors broke the policy of anonymous submission which disqualifies the paper.